# Head-Only Stunning of Turkeys Part 2: Subjective and Objective Assessment of the Application of AC and DC Waveforms

**DOI:** 10.3390/ani11020286

**Published:** 2021-01-23

**Authors:** Steve Wotton, Andrew Grist, Mike O’Callaghan, Ed van Klink

**Affiliations:** Bristol Veterinary School, University of Bristol, BS40 5DU Bristol, UK; Steve.Wotton@bristol.ac.uk (S.W.); andy.grist@bristol.ac.uk (A.G.); mike31@btinternet.com (M.O.)

**Keywords:** turkeys, head-only electrical stunning, voltages

## Abstract

**Simple Summary:**

In this study, we investigated various types of application of head-only electrical stunning to turkeys, in order to improve the effectiveness of the stunning process. We used recordings of physical responses and of electroencephalograms (EEGs) to assess effectivity of stunning and various waveforms: alternating current (AC) and pulsed direct currents (DC). Depending on the waveforms, voltages between 125 and 250 V produced an effective stun in turkeys in this study.

**Abstract:**

Electrical stunning is likely to remain an important stunning method for turkeys at slaughter. The purpose of this study is to understand the application of various waveforms of alternating current (AC) and pulsed direct currents (DC), head-only, to turkeys and to improve the effectiveness of handheld stunning of turkeys. We evaluated the effectiveness of stunning by documenting physical responses and recording electroencephalograms (EEGs). For the assessment of physical responses, the stunning voltage was varied depending on the proportion of animals effectively stunned at a certain voltage level. If all turkeys in a group of 10 were stunned, the voltage was decreased, and the next group was stunned. This was repeated until not all turkeys showed signs of being effectively stunned. The experiment was then repeated at the voltage level just above the one that showed incomplete effective stunning. The effects of the stunning on the EEG recording was assessed in 16 turkeys to measure the occurrence of epileptiform EEGs, in 14 turkeys to assess epileptiform EEGs after neck-cut (bleeding), and in 14 turkeys to assess the effect of increased voltage and reduced frequency on epileptiform EEGs. Assessing EEGs in a laboratory setting contributes considerably to the understanding of electrical stunning procedures. Voltages between 125 and 250 V, depending on the waveform assessed, were effective in producing an effective stun in turkeys in this study.

## 1. Introduction

Current legislation on welfare of animals requires stunning before slaughter to render the animals unconscious, with a few exceptions such as religious slaughter practices [1,2]. Electrical stunning is predominantly used in small scale turkey slaughter [3], while mechanical or controlled atmosphere stunning methods are generally used by larger producers [4,5]. Although controlled atmosphere (CA) killing systems for poultry seem to be gaining ground, investment required for CA systems can often not be justified because of the fact that turkey processing is often seasonal. In 2013, 13,646,425 turkeys were slaughtered in 35 establishments in the UK, with 14 of those establishments processing for less than three months. Consequently, electrical systems are likely to remain in use for some time to come. When the electrical current is applied only to the head, it prevents the direct muscle stimulation produced when it is applied through the whole bird, e.g., using a water bath stunner [6]. With predominantly white fast twitch muscle fibres, the effect of direct muscle stimulation on the production of muscle haemorrhages and broken pectoral bones in water bath stunners will have consequences on turkey meat quality [6,7]. Therefore, the use of a head-only application is likely beneficial to carcass and meat quality. Head only stunning will also be beneficial for animal welfare, due to the fact that the application will usually be less variable than water bath stunning [8,9,10]. Processing properties of turkey meat does not seem to be influenced either positively or negatively by electrical stunning [11,12].

It is important that the stun is applied effectively. In an earlier work [13], a minimum voltage level was established, at which the impedance in the animal tissue of the head was reduced maximally. For different applications, the minimum voltage to overcome impedance was established with either an alternating current (AC) with various frequencies or a direct current (DC) with various duty cycle frequencies [13]. In alternating current applications, 212 V (calibrated as root mean squared (0.707) times the peak voltage) was the voltage level at which impedance the levelled off and was no longer reducing [13]. Knowing the minimum voltage to overcome impedance is, however, not sufficient; the first study was carried out on turkeys that had already been stunned through the application of controlled atmosphere stunning. Therefore, it needs to be established whether effective stuns are actually produced at appropriate voltages. A second study is needed to assess if effective stuns occur, if applied to turkeys that have not been pre-stunned.

This study seeks to increase the understanding of the application of alternating currents (AC) and pulsed direct currents (DC) using hand-held electrodes, head-only, on turkeys. It sets out to improve the effectiveness of hand-held stunning of turkeys by identifying the electrical parameters necessary for an effective stun to enable the industry to reduce the variability seen with current practice. The contribution to the effectiveness of the stun was investigated through pulse duration, duty cycle, applied voltage, current and frequency with pulsed DC waveforms, as well as interactions of these. The effectiveness of stunning was evaluated by (A) documenting physical responses in combination with (B) electroencephalograms (EEGs).

This study was carried out as an empirical study, rather than as an analytical study in which results are assessed statistically. The reason for this is that setting up a statistically sound study design would require larger numbers of birds undergoing incomplete stuns to be able to compare between treatments. This was deemed unethical and therefore a design was used in which at the first sign of a certain application not being effective, the application at that level was stopped.

## 2. Materials and Methods

### 2.1. General Aspects

The study was conducted under Home Office Project License Number PPL 30/2438 for stunning, slaughter, and killing of poultry species, with personal license number PIL 30/1362 (Dr. M. Raj).

A total of 284 turkeys were used for this study.

A prototype mains-powered generator was designed and built for the purpose of this study. The prototype was necessary because at the time of these experiments there was no commercial equipment that could produce the pulsed DC signals that we wanted to test. The device can produce a constant voltage pulsed DC that enabled the following parameters to be controlled:pulse width;duty cycle;applied voltage; andpulse recurring frequency (prf).

Voltage and current values were measured as peak values (0-peak) and peak current values at 0.2 s from the start of current application. The current and voltage generated were recorded using a PR 30 (LEM HEME Ltd., Skelmersdale, UK) current probe and a differential voltage probe (MX 9003, Metrix Electronics, Bramley, UK), respectively.

An appropriate commercial hand-held stunning applicator (Karl Schermer GmbH & Co. KG, Ettingen, Germany, hand-held, small animal stunning tong), which was built for rabbits, was identified and tested and met the requirements for a head-only application with turkeys. It was used in combination with the prototype power generator.

Firstly, for the assessment of physical responses, an experimental protocol was developed in order to minimise the number of birds at risk of receiving low voltages insufficient to stun, whilst maintaining confidence in the results. This approach was deemed required, in order to reduce the number of birds being stunned ineffectively as much as possible, considering the necessity to reduce, refine, and replace use of experimental animals wherever possible, as required in European legislation [14]. In Table 1, a hypothetical sequence is illustrated. Starting voltages were chosen at or above the threshold voltage, i.e., 150 volts 50 Hz, AC [13]. This level represents an average level at which the impedance in the tissue of the head is at its maximum reduction, as shown in earlier work [13]. Whenever 10 consecutive birds were effectively stunned, the subsequent 10 would receive a lower voltage. This was repeated until one bird was judged as “un-stunned”, at which point the treatment group ended (in the example illustrated in Table 1, this was 110 volts). A further 10 birds were then stunned with the voltage of the group immediately before (in Table 1, 125 volts) to verify the effectiveness of the stun at that minimum threshold voltage. AC voltages are given as root mean square (RMS, 0.707 × peak voltage).

### 2.2. Study A

In the first study, turkeys (live weight 16–18 kg, female, age around 15 weeks) were restrained in a shackle suspended from an A-frame at a commercial processing plant. The use of a shackle to restrain individual birds permitted easy access to the head and allowed for subjective evaluation of the bird’s response to the applied voltage. The selected voltage was applied for one second and, provided all birds were stunned, the voltage was reduced for the next group of birds, as described. A one second application time was selected to test whether the applied voltage was sufficient to produce an effective stun “immediately” [15,16,17]. All stun/recovery trials were assessed subjectively by the principal author to prevent observer bias and videotaped for further assessment. Stuns were considered effective if the bird showed signs of initial neck stiffness, absence of rhythmic breathing, absence of spontaneous blinking and third eyelid reflex, absence of other eye reflexes, if it had its eyes open and a fixed gaze, and if an initial tonic phase was followed by a clonic phase [18]. Following the voltage application and assessment, birds were removed from the shackle and processed normally. Every bird for which it was necessary, was immediately re-stunned properly before being processed.

The assessment included:presence of immediate wing flapping (spinal reflexes);time to return of rhythmic breathing (return of functional brain stem); andtime to return of neck tension (later stage in recovery).

A total of 240 individual turkeys were stunned, and the following wave forms and frequencies were used to apply the voltages [10]:AC, 50 Hz frequency, voltage calibrated as root mean square (RMS, 0.707 × peak voltage) voltage from 50 to 300 volts in 25 volt steps (33 birds in total);AC, 200 Hz frequency, voltage calibrated as root mean square (RMS, 0.707 × peak voltage) voltage from 50 to 300 volts in 25 volt steps (35 birds in total);Pulsed DC, 50 Hz frequency, 30% duty cycle, voltage calibrated by peak voltage and applied from 50 to 300 volts in 25 volt steps (31 birds in total);Pulsed DC, 50 Hz frequency, 50% duty cycle, voltage calibrated by peak voltage and applied from 50 to 300 volts in 25 volt steps (33 birds in total);Pulsed DC, 200 Hz frequency, 30% duty cycle, voltage calibrated by peak voltage and applied from 50 to 300 volts in 25 volt steps (32 birds in total);Pulsed DC, 200 Hz frequency, 50% duty cycle, voltage calibrated by peak voltage and applied from 50 to 300 volts in 25 volt steps (33 birds in total);Pulsed DC single voltage spike, 400 µs duration, at 50 Hz frequency, voltage calibrated by peak voltage applied at 50 to 300 volts (39 birds in total); andPulsed DC single voltage spike, 400 µs duration, at 200 Hz frequency, voltage calibrated by peak voltage applied at 50 to 300 volts (4 birds in total).

### 2.3. Study B

In the second study, the effectiveness of stunning was evaluated using electroencephalograms (EEGs). This was applied previously for identification of effective stunning of chickens and for the development of the assessment criteria [15,16,17]. In this study with turkeys, the waveform AC 150 V (RMS) at 200 Hz (identified in earlier work as representing the voltage level providing effective reduction of the impedance of the tissues of the head [13]) was evaluated for the following three criteria to ascertain the effectiveness of the stunning treatment:The occurrence of epileptiform activity in the EEGs following stunning, determined subjectively by examining EEG traces for polyspike activity;Reduction in the total power of the EEG to <10% of pre-stun level following epileptiform activity in three consecutive one second epochs, indicative of a quiescent EEG;During the manifestation of epileptiform and quiescent EEG, birds are considered unconscious and insensible.

The EEG signals were recorded by amplifying them using a pre-amp (10,000×), 0.5–100 Hz filter band (Gould Electric, Gould Nicolet Technologies, Loughton, Essex, UK), after which the amplified EEG signals were passed through a noise removal system (Humbug 50/60 noise removal system: Gould Electric, Gould Nicolet Technologies, Loughton, Essex, UK) to eliminate background noise. They were subsequently digitally recorded at a sampling rate of 20,000 s^−1^ using a vision data acquisition system (Gould Nicolet Technologies, Loughton, Essex, UK) [15].

This study consisted of three parts:The effectiveness of the applied stun was evaluated using EEGs. The ability of this waveform, applied head-only for one second, to induce epileptiform activity and abolish somatosensory evoked potentials (SEPs) in 16 turkeys was evaluated in the laboratory.

The turkeys were sedated and anaesthetised using ketamine hydrochloride and pentobarbitone sodium. They were then implanted with silver-silver chloride EEG recording electrodes on the surface of the dura mater and somatosensory stimulating electrodes [19]. After overnight recovery, the birds were restrained on a shackle, further restrained with adhesive tape to prevent wing flapping, and stunned head-only for 1 s. SEP’s were evoked by electrical stimulation (0.5–1 volt, 2 ms pulse, 2 stimuli per second) of the contralateral radial nerve at a stimulus rate of 2 stimuli per second and recorded digitally for about two minutes before and up to two minutes after the stunning treatment [15]. The period of two minutes before until two minutes after the stun was necessary to produce an average SEP to demonstrate that it was abolished by an effective stun. Earlier work suggested that the pre-stun SER stimulation was not particularly painful to the birds [15]. The birds were humanely killed on completion of the recording.II.To assess the effect of the stun in combination with normal slaughter processing, i.e., bleeding to cause the death of the animal, 14 turkeys were implanted with EEG electrodes in the same manner as in the first part of the experiment, and sinusoidal AC at 200 Hz and 150 V (RMS), was applied for 6 s (SD ± 1.24), followed by an effective ventral neck cut within 9.86 s (SD ± 1.247). SEP’s were evoked in these turkeys by electrical stimulation (0.5–1 volt pulses) of the contralateral radial nerve at a stimulus rate of 2 stimuli per second and recorded digitally for about two minutes before and up to the induction of an isoelectric EEG following stunning and slaughter [15]. After completion of the recording, these birds were killed humanely.III.To evaluate the effect of varying voltage/frequency combinations on the effectiveness of the stun on the basis of the EEG, 14 turkeys, after EEG electrodes having been implanted in the same manner, were stunned using a variety of voltage/frequency combinations. These birds were also humanely killed after completion of the experiment.

The number of birds included in study B had to be kept to a minimum because of the welfare implications of the procedure and treatment of the animals, as well as the limits of the licenses used. For that reason, only a very limited scope of voltage and frequency variations could be included in the assessment. The authors have, on the basis of their experience and earlier work [13], decided which voltage/frequency combinations were expected to be used in practice most often and have included these in the assessment.

These assessments, together with an overall impression of the effect of each treatment on each bird, enabled a minimum threshold voltage to be determined for effective head-only stunning of turkeys.

Analysis of the effect of the head-only stun on SEP’s was inconclusive due to their indefinite size and were therefore not used to interpret the effectiveness of the stunning treatments.

### 2.4. Statistics

The results of the study are given as descriptive statistics in the tables provided. In study A, any treatment was immediately stopped if any animal received an ineffective stun, as determined subjectively by the principal author and if the treatment at that voltage level was deemed unacceptable. For each treatment, the conclusion was made that the voltage level immediately above the one where stuns were ineffective were acceptable. In study B, only a limited number of specific treatments were chosen to be included based on experience and knowledge of the authors.

## 3. Results

In Table 2, the results are given for the assessment of physical responses at each voltage–current combination and waveform. In all AC waveforms evaluated, both 150 and 125 volts (RMS) produced effective stuns. Below 125 volts, the stuns were insufficiently effective. In the DC waveform groups, DC 50 Hz with 30% duty cycle and DC 50 Hz with 50% duty cycle stuns were also effective in the two higher voltage levels. The same applies for the DC applications at 200 Hz and the DC Spike application at 200 Hz. Spike application at 50 Hz did not produce an effective stun at any level.

Table 3 shows the number of successfully stunned birds across the stunning groups in relation to the applied voltage. The values at which stuns are effective can be recognised again, as in Table 2, and this table also shows that at a direct current applied at 50 Hz with 50% duty cycle seems to produce less effective stuns compared to when it is applied with a 30% duty cycle. This effect is not present with application of DC at 200 Hz.

In Table 4, the average time for the return of physical reflexes following the different stunning treatments is represented. Return of rhythmic breathing is usually the first sign of regaining consciousness. In all cases where stuns were considered to be effective, return of rhythmic breathing did not occur earlier than at least 20 s, with a fairly small confidence interval. Head righting did not occur before at least 30 s, and in most cases, well beyond that. There is some variation between the treatment groups, though in most cases the time that lapsed for return of rhythmic breathing and head righting was in the same range. When a treatment group contained birds that were not stunned effectively, the return of head righting tended to be a little shorter.

The results for the DC 200 Hz group for both 50% and 30% duty cycles did not fit the overall trend for a rise in the threshold voltage necessary for an effective stun across the treatment groups. It is not known whether these lower thresholds for effective stuns (125 volts for both waveforms) are a physical effect of this frequency and waveform with turkeys, or an anomaly of the experimental method.

The justification for recording the RMS current at 200 ms (Table 2) was based on research evidence with electrical head-only stunning of sheep as follows:Successful electrical stunning requires a minimum of 1.0 Amp in sheep (EC Regulation 1099/2009).However, 1.0 Amp applied for less than 200 ms in sheep will not produce epileptiform activity in the brain [20].Therefore, the electrical criteria for an immediate stun with sheep is ≥1.0 Amp within 200 ms.

Brain dysfunction will take place within this time period as the effect of a swinging current will interrupt the biological function of neuronal membranes.

Criteria were chosen that indicated whether sufficient current was applied to induce an effective stun [15,16,17], as described in materials and methods.

An example of the application of these criteria is shown in Figure 1.

The results of the EEG assessment of the ability of each waveform, applied head-only for one second, to induce epileptiform activity are presented in Table 5. Remarkably, no typical polyspike activity was seen in any of the animals. In most cases, the reduced EEG activity (less than 10% of the pre-stun EEG activity) lasted more than a minute after the application of the stun.

The results of the application of current for 6 s (SD ± 1.24) followed by an effective ventral neck cut within 9.86 s (SD ± 1.247) on the EEG are presented in Table 6. Here, the effect of the stun was combined with the effect of the neck cut to exsanguinate the bird. As in the earlier experiment, clear polyspikes were not seen. The EEG showed reduced activity below 10% of pre-stun levels within 10 to 40 s after application of the stun and the neck cut in all birds.

The results of the determination of an effective stunning voltage/frequency combination on the basis of the occurrence of epileptiform EEGs are presented in Table 7. In this experiment, polyspike activity was seen in several of the birds. There seems to be a limited relationship between voltage and the period of reduced EEG activity, though at the voltages around 250 V RMS, the period of reduced activity seems to be of longer duration.

Figure 1a shows an example of the EEG recording (turkey no. 6 (Table 7)), pre- and post-stun with 200 volts, sine wave AC at 50 Hz; Figure 1b shows the effect of the stunning on % change in the total power content (FFT; Fast Fourier transform) of the EEG [15]. The EEG showed polyspike activity (3–8 Hz) followed by suppression to below 10% of pre-stun values between 13 and 28 s from the end of stunning current application. Therefore, turkey 6 met the required criteria for effective electrical stunning [15,16,17]. Figure 1c,d show the voltage and current profiles for the head-only stunning application with this bird. The initial build-up in current shown in the stun current profile (Figure 1d) reflects the inherent high resistance of living tissue that breaks down with the applied voltage and time [21].Figure 1The EEG, FFT(Fast Fourier transform, an indicator of total power content) analysis, voltage, and current profiles from Turkey #6 from Table 7.
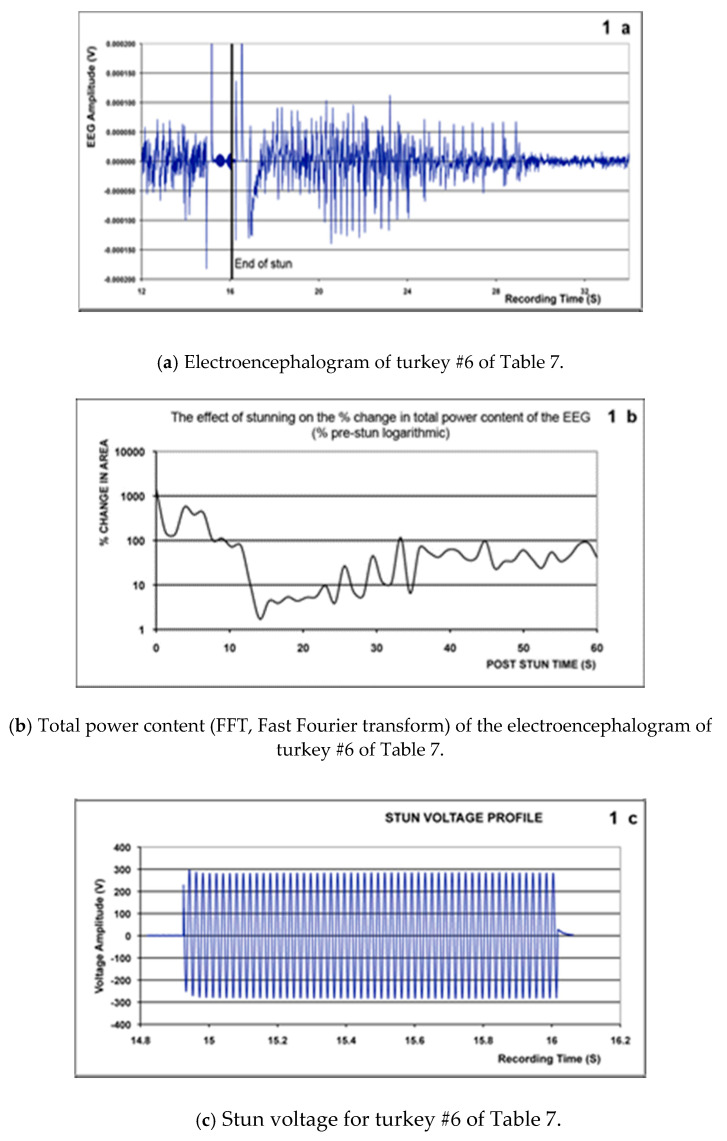

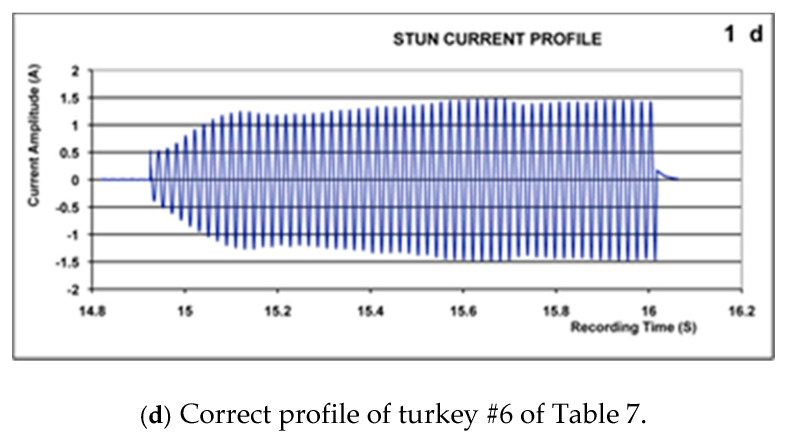


## 4. Discussion

The subjective assessment of the physical response of turkeys to electrical head-only stunning is probably more reliable than the assessment following water bath stunning, as the characteristics of the application are likely to be better controlled because each bird is assessed individually in its turn. There is currently considerable debate among scientists about the reliability of criteria used to determine effective stunning when the current is applied to the whole bird in a water bath. The current could affect peripheral nerves and induce muscle paralysis in a bird that remains conscious. This is not the case with head-only applications, and the results of Table 4 clearly show that the time for return of neck tension tended to be reduced in those treatment groups that contained an un-stunned bird, whereas the time for the onset of rhythmic breathing was not so decisive. This may have been affected by the extensive physical activity (violent spasm and strong flapping movements) that made the identification of the return to rhythmic breathing difficult in practice. A short application of stunning current that is of sufficient magnitude to result in an effective stun will result in extensive physical activity due to spinal reflexes in the higher centres of the brain. When a head-only stun is applied for a longer period, e.g., 6 s, the spreading current field will result in spinal deactivation and subsequently less physical activity [22]. The stun duration, it should be mentioned, does not imply the effect of the stunning is not immediate upon the start of application. As can be seen from Table 5 and 6, where stunning variables produced an effective stun, both treatments reflect the effectivity of the stun in the EEG.

The starting voltages, when using single voltage spike application, were considerably higher than for the other waveforms (Table 3 and 4). This starting level was chosen on the basis of earlier work [13] and proved to be necessary, as these applications would not have produced effective stuns at lower voltages.

In Table 8, the results were compared to minimum voltage recommendations made in an earlier paper by the same authors [13] for the waveforms and frequencies studied.

When the resulting currents from these voltages were compared to a previously published recommended minimum current of 400 mA [19], a number of the recommendations produced current levels below 400 mA. These differences may be a result of:Different waveforms applied. Previous studies have assessed the effect of 50 Hz sine wave AC on head-only stunning of turkeys [19].Different methods of assessment (subjective vs. objective).The larger number of birds being used in this study. The accuracy of this subjective assessment to predict the minimum voltage required for stunning could be compared to the evaluation of effectiveness using EEGs.

Anecdotally, the authors observed that the position of the stunning tong electrodes appeared to be significant in the production of epileptiform activity in the EEGs. When the electrodes were placed over the eyes, the resulting RMS current was relatively high, but the application was usually ineffective. However, when the electrodes were placed behind the eyes, the RMS current was reduced but the application was generally more effective.

Polyspike activity was only observed in a minority of the turkeys. Polyspikes are seen at the start of epileptic seizures and absence seizures [23,24]. We would have expected to see them in most animals, as stunning is considered to be the result of an epileptic fit [25,26]. A relevant aspect may be that while generalised epilepsy following electrical stunning can generally be recognised in animals through the presence of synchronised 8 to 13 Hz electrical activity in the EEG, there are indications that low frequency epileptiform activity occurs in poultry [27].

The reduction of the frequency of the stunning current waveform for 50 Hz in combination with an increase in the voltage to 200 volts resulted in an effective stun in 100% of the turkeys (*n* = 8). Interestingly, the application of 250 volts at 50 Hz (*n* = 2) was less effective, which could be due to the positioning of the stunning tongs or its contact with the turkey heads. It is suggested that further research is necessary to determine whether a refined electrode shape together with an electrode application in a position that reflects the anatomy of the head (behind the eyes) would be more successful with lower voltages. The comparison of a subjective assessment of an effective head-only stun with turkeys [13] with an objective assessment using implanted EEG electrodes and the criteria outlined above adds further evidence to the proposal that laboratory testing should be used to determine the voltage required for effective stunning for different waveforms and frequencies, and that subjective assessment in the field is inaccurate.

## 5. Conclusions

Various levels of voltage were shown to provide effective stuns in turkeys. Subjective assessment of stunning results combined with EEGs performed under laboratory circumstances is a good method to assess effectiveness of stunning. Further research into optimising the electrode shape may be helpful in improving stunning results.

## Figures and Tables

**Table 1 animals-11-00286-t001:** Protocol for 10–20 turkeys per voltage treatment group, based on hypothetical results.

Applied Voltage AC 50 Hz	% Stunned (*n*—Turkeys)	Predicted 95% CI Range
150	100% (10)	72–100%
140	100% (10)	72–100%
125	100% (20, in two batches of 10)	84–100%
110	80% (8)	38–96%

**Table 2 animals-11-00286-t002:** Mean average current values for the first 200 ms and the effectiveness of head-only stunning, subjectively assessed.

AC ^1^ Groups	V (RMS ^2^)	*n*	RMS Current at 0.2 s from Stun Start (A)	Birds Effectively Stunned %
AC 50 Hz	150	10	0.50	100
	125	20	0.53	100
	100	3	0.48	67
AC 200 Hz	150	10	0.87	100
	125	20	0.52	100
	100	5	0.34	80
Pulsed DC ^3^ Groups	DC volts (average)	*n*	Average current at 0.2 s from stun start (A)	Birds effectively stunned %
DC 50 Hz with 30% duty cycle	53	10	0.23	100
	45	20	0.15	100
	38	1	0.24	0
DC 50 Hz with 50% duty cycle	100	10	0.47	100
	88	20	0.42	100
	75	3	0.36	67
DC 200 Hz with 30% duty cycle	45	10	0.14	100
	38	20	0.12	100
	30	2	0.07	50
DC 200 Hz with 50% duty cycle	75	10	0.21	100
	63	20	0.21	100
	50	3	0.10	67
SP ^4^ 200 Hz	24	10	0.02	100
	22	20	0.02	100
	20	7	0.02	86
	18	2	0.01	50
SP 50 Hz	6	1	0.02	0
	5.5	1	0.03	0
	5	1	0.02	0
	4.5	1	0.01	0

^1^ AC = alternating current. ^2^ RMS = root mean square (0.707 × peak voltage). ^3^ DC = direct current. ^4^ SP = pulsed DC single voltage spike.

**Table 3 animals-11-00286-t003:** Treatment groups showing the number of successful head-only stunned turkeys.

		Number of Successful Stuns/Number of Turkeys								
	Voltages (RMS ^2^)	100	125	150	175	200	225	250	275	300
Groups										
AC ^1^ 50 Hz		2/3	20/20	10/10						
AC 200 Hz		4/5	20/20	10/10						
DC ^3^ 50 Hz with 30% duty cycle			0/1	20/20	10/10					
DC 50 Hz with 50% duty cycle				2/3	20/20	10/10				
DC 200 Hz with 30%duty cycle		1/2	20/20	10/10						
DC 200 Hz with 50% duty cycle		2/3	20/20	10/10						
SP ^4^ 200 Hz							1/2	6/7	20/20	10/10
SP 50 Hz							0/1	0/1	0/1	0/1

^1^ AC = alternating current. ^2^ RMS = root mean square (0.707 × peak voltage). ^3^ DC = direct current. ^4^ SP = pulsed DC single voltage spike.

**Table 4 animals-11-00286-t004:** Average times (s) for return of spontaneous physical reflexes following 1 s application of head-only stunning in turkeys.

Group	V		Return of Rhythmic Breathing (s)	Return of Head Righting (s)	Animals Stunned
		*n*	Mean ± SD	Mean ± SD	%
AC ^1^ 50 Hz	150	10	29 ± 3.81	67 ± 16.79	100
	125	20	26 ± 2.83	50 ± 21.00	100
	100	3	23 ± 3.21	33 ± 2.12	67
AC 200 Hz	150	10	26 ± 3.84	49 ± 15.97	100
	125	20	26 ± 4.25	44 ± 12.70	100
	100	5	21 ± 3.87	41 ± 13.58	80
DC ^2^ 50 Hz with 30% duty cycle	175	10	25 ± 3.59	42 ± 14.48	100
	150	20	25 ± 2.66	39 ± 14.49	100
	125	1	15		0
DC 50 Hz with 50% duty cycle	200	10	26 ± 3.56	41 ± 12.59	100
	175	20	25 ± 3.36	49 ± 13.15	100
	150	3	25 ± 2.65	36 ± 1.53	67
DC 200 Hz with 30% duty cycle	150	10	25 ± 2.63	44 ± 13.04	100
	125	20	26 ± 3.09	40 ± 10.93	100
	100	2	17		50
DC 200 Hz with 50% duty cycle	150	10	24 ± 3.20	42 ± 10.99	100
	125	20	24 ± 3.72	39 ± 9.55	100
	100	3	23 ± 7.78	66	67
SP ^3^ 200 Hz	300	10	27 ± 3.63	38 ± 6.32	100
	275	20	27 ± 2.62	38 ± 7.51	100
	250	7	27 ± 3.96	41 ± 9.62	86
	225	2	26	32	50
SP 50 Hz	300	1	25	35	0
	275	1			0
	250	1			0
	225	1			0

^1^ AC = alternating current, given in RMS (root mean square (0.707 × peak voltage). ^2^ DC = direct current. ^3^ SP = pulsed DC single voltage spike.

**Table 5 animals-11-00286-t005:** The effect of head-only stunning on the expression of epileptiform electroencephalograms (EEGs) in 16 turkeys.

*n*	Head-Only Stunning Parameters				EEG Characteristics		Dead Weight (kg)
	Frequency (Hz)	RMS ^1^ Voltage (volts)	RMS Current (mA)	Stun Time (s)	Presence of Polyspike Activity (0/1)	Post Polyspike <10% Pre-Stun (0/1) (s)	
1	200	147.9	819	1	0	1 (>20)	5.9
2	200	148.0	797	1	0	1 (>120)	5.3
3	200	148.2	918	1	0	0	4.5
4	200	148.6	1049	1	0	1 (0–120)	7.4
5	200	148.2	737	1	0	1 (0–10)	5.6
6	200	148.3	898	1	0	1 (30–120)	6.0
7	200	148.4	904	1	0	1 (25–100)	5.6
8	200	148.5	963	1	0	1 (2–20, 56–120)	5.5
9	200	148.4	1007	1	0	1 (22–120)	5.7
10	200	148.4	1070	1	0	1 (5–120)	5.4
11	200	148.9	957	1	0	1 (2–100)	7.3
12	200	148.3	688	1	0	1 (25–90)	5.3
14	200	148.7	965	1	0	1 (1–120)	5.4
15	200	148.3	388	1	0	1 (20–120)	6.0
16	200	148.5	760	1	0	1 (40–120)	7.0

^1^ Root mean square (0.707 × peak voltage).

**Table 6 animals-11-00286-t006:** The effect of head-only stunning followed by slaughter on the electroencephalogram (EEG) in turkeys.

*n*	Head-Only Stunning Parameters				Stun-Neck Cut (s)	EEG Characteristics		Dead Wt. (kg)
	Frequency (Hz)	RMS ^1^ Voltage (Volts)	RMS Current (mA)	Stun Time (s)		Presence of Polyspike Activity (0/1)	Time to Onset of <10% Pre-Stun Total Power (s)	
1	200	149.2	1293	6.7	10	0	>15	6.6
2	200	149.1	1394	6.7	11.5	0	>40	4.3
3	200	149.2	1026	6.7	8.2	0	>10	5.5
4	200	149.2	1096	6.7	9.1	0	>15	5.6
5	200	149.3	1069	6.7	9.1	0	>25	6.6
6	200	149.1	823	6.7	9.4	0	>40	
7	200	149.4	1164	3.9	9.1	0	>20	4.8
8	200	149.1	1027	6.7	9.4	0	>15	4
9	200	149.2	809	3.4	11.5	0	0	5.5
10	200	149.1	1097	6.7	10.6	0	>13	5.4
11	200	149.2	847	4.4	11.8	0	>35	5.8
12	200	149.2	758	6.7	7.3	0	>20	5.5
13	200	149	1040	4.5	10.3	0	>15	6.1
14	200	149	1419	6.7	10.3	0	0	5

^1^ Root mean square (0.707 × peak voltage).

**Table 7 animals-11-00286-t007:** The effect of increased voltage and reduced frequency on the induction of epileptiform activity in the electroencephalogram (EEG) in turkeys.

*n*	Head-Only Stunning Parameters				EEG Characteristics		Dead Wt.(kg)
	Frequency (Hz)	RMS ^1^ Voltage (volts)	RMS Current (mA)	Stun Time (s)	Presence of Polyspike Activity (0/1)	Post Polyspike <10% Pre-Stun (0/1) (s)	
1	50	111.3	473	1	1	0	9.6
2	50	150.7	858	1	1	0	10.4
3	50	150.7	434	1	1	0	9.1
4	200	199.6	1313	1	0	1 (0–60)	10.9
5	50	198.1	1209	1	(0–12 s lost)	1 (0–50)	8.1
6	50	197.8	888	1	1	1 (13–28)	9.6
7	50	201.3	967	1	0	1 (20–38)	10.8
8	50	199.6	882	1	1	1 (18–35)	9.7
9	50	199.1	900	1	1	1 (22–40)	9.4
10	50	199	711	1	1	1 (10–16)	9.4
11	50	199.9	863	1	1	1 (10–34)	11.7
12	50	199.8	723	1	1	1 (14–38)	9.3
13	50	249.7	1741	1	0	1 (3–120)	12.8
14	50	249.7	1582	1	0	1 (2–120)	11.2

^1^ Root mean square (0.707 × peak voltage). Differences in colour indicate different frequency-voltage combinations.

**Table 8 animals-11-00286-t008:** Comparison of the recommended voltages for an effective stun.

Treatment Group		This Study (Volts RMS)	Wotton et al., 2020 [13] (Volts RMS)
AC	50 Hz	125	150
AC	200	125	150
DC	50 Hz at 30% duty cycle	150	175
DC	50 Hz at 50% duty cycle	175	150
DC	200 Hz at 30% duty cycle	125	150
DC	200 Hz at 50% duty cycle	125	150
400 µs DC	50 Hz Voltage Spike	250	225
400 μs DC	200 Hz Voltage Spike	-*	225

* No effective stun seen.

## Data Availability

The data presented in this study are available on request from the corresponding author.

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
