# Peer review of "Head-Only Stunning of Turkeys Part 2: Subjective and Objective Assessment of the Application of AC and DC Waveforms"

_animals, 2021, doi:10.3390/ani11020286_

Round 1

Reviewer 1 Report

Dear editors,

dear authors,

After revision by the authors, everything is comprehensible and well implemented.

Now, the material and method of this study is clear organized and the description of the results are fin. I have no further comments.

Kind regards.

Author Response

Thank you very much for your favourable remarks.

Reviewer 2 Report

I believe that revision (and re-evaluation) of this manuscript is still required to address the following issues:
1. Introduction: Lines 55-61 refer to previous publication. This paragraph should be extended and improved to clearly present what has been already done on the subject issue and explain why the current study consists 'part 2' in the title. Also, in this section you should state that this is an empirical study (not an analytical one).
2. Material and Methods: Ethical Committee approval number should be added here. The number of animals used should be explained as related to 3Rs. Statistical analysis paragraph should be deleted (not acceptable in current form). Instead you should provide a paragraph under the subtitle 'Statistics' and refer to the descriptive statistics methods you applied to summarize your data.
3. More references should be added from the available literature.

Author Response

Thank you for your remarks. We have adjusted the text on the basis of your remarks as follows:

  1. We have extended the text of the paragraph of lines 55 -61 as follows to explain more about the results of part 1: “It is important that the stun is applied effectively. In earlier work (10) a minimum voltage level has been established at which the impedance in the animal tissue of the head is reduced maximally. For different applications, either alternating current (AC) with various frequencies or direct current (DC) with various duty cycle frequencies, the minimum voltage to overcome impedance could be established (10). In alternating current applications, 212 Volt (calibrated as root mean squared (0.707) times the peak voltage) was the voltage level at which impedance was levelling off and was no longer reducing (10). Knowing the minimum voltage to overcome impedance is however not sufficient; the first study was carried out on turkeys that had already been stunned through the application of controlled atmosphere stunning. Therefore it needs to be established whether at appropriate voltages effective stuns are actually produced. A second study is needed to assess if effective stuns occur, if applied to turkeys that have not be pre-stunned.”

We have also added some lines about the fact that this is an empirical rather than an analytical study: lines 74-78.

  1. The Ethical Committee approval number, which was in the acknowledgements in the previous version, has been taken to the first paragraph of the Materials and Methods, lines 82-83.

We have added a reference to the 3R’s in line 108-111.

We have replaced the text in the Statistics paragraph, hoping that the current version provides a clearer understanding of the approach chosen to arrive at our conclusions.

  1. We have added a few more relevant references.

Reviewer 3 Report

Thanks, to the authors. All my questions are answer.

I only have one aditional question left and if the authors can answer this than it's okay for me to publish this artical.

In your artical the is no information given on ethical clearance. Ethical clearance is to receive a green light from an ethics committee to proceed with the research. This information is relevant as national legislation in the UK set standards for experiments with live animals. The code of their license has to be given in the Materials and Methods.

Author Response

Thank you for your remarks. As to the ethical clearance: we had included the license numbers we used in the Acknowledgements in the previous version. We have now included them in the Materials and Methods.

Round 2

Reviewer 2 Report

The revised manuscript can now be acceptable for publication by Animals Journal.

This manuscript is a resubmission of an earlier submission. The following is a list of the peer review reports and author responses from that submission.

Round 1

Reviewer 1 Report

In the introduction you write about better meat quality and less broken bones by using head only stunning. Of course this is very important. But it is well known that head only stunning improves animal welfare. Stunning in a waterbath with 12 to 24 animals gives a wide variation of Amperages for each turkey.

Can you describe how EEG was recorded, which software program did you use?

In table 2 you used "SP 200 Hz" please can you describe what "SP" means?

How were animals treated after stunning?

table 4 why such a big jump in voltage at SP 200Hz and SP 50 Hz?

figure 1 the numbers are not very clear.

    ot very clear.

Author Response

Thank you for your comments. It is absolutely true that head-only stunning can be applied more consistently and therefore with greater assurance that the stun is successful. We have added the following text in line 52 (new version): "Head only stunning will also be beneficial to animal health due to the fact that application will usually be less variable than water bath stunning (7, 8, 9)."

The EEG signals were amplified using a pre-amp (10 000 x), 0.5–100 Hz filter band; Gould Electric, Gould Nicolet Technologies, Loughton, Essex, UK). The amplified EEG signals were passed through a noise removal system (Humbug 50/60 noise removal system: Gould Electric, Gould Nicolet Technologies, Loughton, Essex, UK) to eliminate background noise and were then digitally recorded  at a sampling rate of 20 000 s–1  using a Vision Data Acquisition System (Gould Nicolet Technologies, Loughton, Essex, UK). We have added this information in the text, lines: 163-168

SP in this context means DC Spike application of voltage. We realise that we have not listed the treatments very clearly. We have now corrected that in lines: 130-147

In the first experiment, after stunning the animals that were successfully stunned went into the normal processing (see line 122-123). In the second experiment the animals were, where applicable, killed as a result of the stun or humanely killed as soon as possible.

There is indeed a big jump in voltage when applying spike voltages. The reason we did this, is that on the basis of earlier work, it was known that these applications would only be effective at higher voltages. We have added some background in the discussion in lines: 353-356

Unfortunately, we do not have alternative pictures for figure 1.

Reviewer 2 Report

This study, although on an interesting scientific topic, does not comply with the standards of a scientific publication.

Statistical design is missing and all results obtained cannot be supported under statistical uncertainty and probabilities.

There is no approval by Ethical Committee on this very sensitive issue of animal killing methods.

Introduction is not sufficient, references are few, some justifications are based on sheep indicators.

The study can be characterized as an empirical and pilot study and should be re-submitted for evaluation as a short communication.

Author Response

Thank you for your comments.

We have made adjustments in order to improve the text of the paper. Hopefully it can meet your approval upon second review.

We have indeed not done any statistics on the data. The reason for this is, that we could not use an unlimited number of animals, particularly for the EEG studies. This is in relation to the welfare and ethics issues involved. As a result, the power of any statistical analysis would be too low, and as a result would not provide better results than the observational data that we produced. We acknowledge that the results mainly have practical use, and this is the reason for the design of the slaughter line stunning design (Study A in the new version), where the voltage level combination that provided an effective stun just above the one that did not, was repeated to confirm its efficacy. We have added an explanation on the absence of statistical analysis in lines 219-224.

We did have ethical approval under the licenses Home Office Project Licence Number PPL 30/2438 Stunning, Slaughter and Killing of Poultry Species and Personal Licence Number PIL 30/1362. This information has now been added in the acknowledgements in lines 418-419.

We have added information to the introduction and hope that that has improved the introduction.

We agree that the study can be characterised as an empirical study. We do not agree that it should be seen as a pilot study. It is a practical study, without a doubt, but the particulars of the subject, i.e. stunning of animals, and therefore the necessity to make do with limited numbers of subjects in the study, makes it difficult to carry out these studies in another way. Whether the study should be re-submitted as a short communication, we would like to leave to the judgement of the Editor.

Reviewer 3 Report

Review: Title: Head-only Stunning of Turkeys 2: Subjective and objective assessment
of the application of AC and DC waveforms.

Comments

To the editor:

General Comments

The study deals with the effect of various physical/electrical parameters of electrical head-only Stunning of Turkeys. The investigation was done with two different waveforms and a total number of 284 turkeys. In general, the article discusses an important and interesting research theme. About the language, the article is well written and I did not notice grammatical errors.

The introduction is well organized but some information about the legal situation of Stunning turkeys are missing (for EU 1099…). Why the authors focused on the alternating current (AC) and pulsed direct currents (DC) is not clear (but I`m no specialist in Stunning methods). The aim of the study is not clear in total.  The material and method of this study is not clear. More information about:-        the animals (male or female)-        was that a registered animal experiment? -        the prototype generator (what was the reason to build a prototype?) and the commercial applicator (is this for turkeys? Negative parameters = what is suboptimal about the applicator?....). The Stunning applicator (“pliers”) is combined with the prototype – is this correct?Study 1: -        The description of the groups (which shown Table 2 in the results) are missing. DC Voltages are missing. I am no specialist, but the authors write about AC but not about DC? -        If you not read the results, the number of 240 birds are not clear-        How was the stunning effect evaluated in study 1? Have clinical sings, such as reflexes, been tested? Interobserver reliability? -        How does it possible to subject the turkeys to normal slaughter process after the observation of stunning and recovery assessment?Study 2-        Why do you test the waveform AC 150 V at 200 Hz and no other Voltes … for EEGs? -        The implantation Technic (119 ff) is used for all three parts? It is only described in part 1.aim of the 3 parts are not clear. What is the reason for the animals used?-        Why SEP´s were evoked 2 minutes before stunning? Is that associated with pain?

  • Data presentation and statistical analysis are missing

   The results are long represented (e.g. 6 tables and figures in the discussion). The description of the tables are some more explanations necessary (for example terms and used abbreviations) The discussion is clear written, nevertheless at some points there are some more explanations /interpretations necessary.

Specific Comments

L40: Write abbreviation: CA = alternating current

L69: Detailed information about the stunning applicator are missing (company, Ettingen Germany)

L142/145: this part at the beginning of the material and methods

L191,192: The table is not well described (for example What does “SP”, “RMS”, “DA”, “AC”  means ; this is also important for the next tables and figures)

Author Response

Thank you for your comments. We will try to answer each comment as well as possible.

We have added some references to stunning of animals prior to slaughter to the introduction, see lines 38-39.

We have focussed on Alternating current (AC) and Direct current (DC), because these are the two ways in which electricity can be provided. Stunning equipment for head-only stunning is available in both forms, so we needed to include both in the study.

The aim of the study is described in the last paragraph of the introduction. We have added a paragraph about the reliability of stunning depending on the electrical characteristics which hopefully clarifies it a bit further: line 55 -62.

We have tried to clarify the materials and methods section by partitioning it more clearly. We have also added some information about the treatment of the animals.

Information about the animals has been added in line 110.

The animal experiment was conducted under licence and was registered. Details about the relevant licences has been added to the Acknowledgements, line 418-419.

We needed to build a prototype because we needed to be able to apply the variable voltages and currents that we needed for the assessment. At the time of these experiments there was no commercial equipment producing the pulsed DC signals that we wanted to test. The commercial applicator was built for rabbits but filled the purpose for turkeys. The stunning applicator was used in combination with the generator. We have clarified that in lines 77-80.

The description of the groups has been added: lines 130-147

We do write about AC AND DC. We hope that adding the description of the groups makes it clearer.

We have added the number of birds per treatment to the materials and methods. We have also followed your suggestion to move (part of) the last paragraph of materials and methods to the top.

We have added the signs used to evaluate the quality of the stun in lines 118 - 121. The principal author was the only observer, to avoid observer bias, see line 117.

The turkeys in the experiment were taken out of the normal shipment to the abattoir, and were in principle stunned like all the other birds, and nothing was done that would impair the quality or safety of the product. Therefore, after, where necessary, corrective stun, the animals could be processed normally.

Study 2

The experiments impact severely on the welfare of the birds subjected to the treatments. Therefore we needed to keep the numbers of birds low and we had to make a choice in what waveforms to study. In earlier work, the AC 150 V/200 Hz was identified as an effective waveform. Therefore we chose to use this waveform as a starting point. We have added some information regarding this choice to the text, lines 153-154 and lines 203-208

The implantation technique was indeed used in all three parts of Study B. We have added references to this in the text. We have also tried to clarify the aim of the three parts, between lines 171-201

SEP’s were evoked from 2 minutes before until 2 minutes after stunning, because it was necessary to produce an average SEP to demonstrate that it was abolished by an effective stun. The pre stun SER stimulation was not particularly painful to the birds. We have added some information about this in the text in lines: 182-184

We have not done statistical analysis, as the numbers of animals involved in each of the categories were of necessity small. Therefore statistical analysis would not have had sufficient power. We needed to rely on interpretation of the observed empirical results to draw conclusion. We have added an explanation in lines 219-224

We have added explanations of the abbreviations in the tables and hope we have improved the readability of the tables with our changes.

Specific Comments

L40 (new version line 43): Write abbreviation: CA = alternating current: in this context CA means Controlled Atmosphere. We have added the acronym to the text in the introduction to make this clearer.

L69: Detailed information about the stunning applicator are missing (company, Ettingen Germany). We have added this information now in line 89-90.

L142/145 (new version lines 214-215): this part at the beginning of the material and methods: We have moved part of this paragraph to the top of material and methods. The last sentence we feel should be kept in this place as it refers to SEP's which are explained just before this.

L191,192: The table is not well described (for example What does “SP”, “RMS”, “DA”, “AC”  means ; this is also important for the next tables and figures): We have adjusted the tables and explained all the abbreviations.